# Indication of critical scaling in time during the relaxation of an open quantum system

Ling-Na Wu [1,2,4], Jens Nettersheim[3,4], Julian Feß[3], Alexander Schnell [1], Sabrina Burgardt[3], Silvia Hiebel[3], Daniel Adam[3], André Eckardt [1]✉ & Artur Widera [3]✉

Near continuous phase transitions, universal power-law scaling, characterized by critical exponents, emerges. This behavior reflects the singular responses of physical systems to continuous control parameters like temperature or external fields. Universal scaling extends to non-equilibrium dynamics in isolated quantum systems after a quench, where time takes the role of the control parameter. Our research unveils critical scaling in time also during the relaxation dynamics of an open quantum system. Here we experimentally realize such a system by the spin of individual Cesium atoms dissipatively coupled through spin-exchange processes to a bath of ultracold Rubidium atoms. Through a finite-size scaling analysis of the entropy dynamics via numerical simulations, we identify a critical point in time in the thermodynamic limit. This critical point is accompanied by the divergence of a characteristic length, which is described by critical exponents that turn out to be unaffected by system specifics.

Phase transitions emerge from the collective behavior of large quantum systems in the thermodynamic limit[1]. A continuous phase transition is signaled by the divergence of a characteristic length scale $\xi$, when the control parameter approaches a critical value. As a result, the behavior near the transition becomes independent of the microscopic details of a system, giving rise to universal critical exponents[1], like the one describing the divergence of $\xi$ as a function of the control parameter. Despite the fact that the distinction between different phases of matter, like liquid or crystalline, is an essential and well-known aspect of nature, phase transitions and their critical behavior remain an active field of research until today. Subjects of interest include, for instance, quantum phase transitions happening in pure quantum ground states at absolute zero[2] and topological phase transitions beyond Landau's paradigm[3].

Recently, the transient evolution of isolated quantum systems gained considerable interest, as it can be realized in engineered quantum systems such as ultracold atomic quantum gases. Prominent effects that were studied include the transition between eigenstate thermalization and many-body localization[4–6], non-equilibrium phase transitions in the long-time (or prethermal) behavior of (almost) integrable quantum systems[7,8], or the observation of discrete time crystals in interacting Floquet systems[9,10]. Another fascinating example is the prediction and observation of dynamical quantum phase transitions[11–15] and universal scaling behavior[16–20] occurring at a critical time during the transient non-equilibrium evolution of isolated quantum systems. Here *time* plays the role of the control parameter. The underlying non-equilibrium dynamics can be initialized, for example, by a quantum quench, i.e., a rapid parameter variation starting from the ground state of the previous Hamiltonian.

In the following, we describe another example of critical behavior with respect to time, reminiscent of a continuous phase transition associated with the dynamics of a quantum system. It happens during the relaxation of an *open* system and corresponds to the divergence of a localization length $\xi$ at a critical time. It is, thus, different from the dynamical quantum phase transitions associated with the non-analytic behavior of the return probability in isolated systems, as they were described previously[15]. Importantly, it is also different compared to non-equilibrium phase transitions occurring in the long-time

[1]Institut für Theoretische Physik, Technische Universität Berlin, Hardenbergstraße 36, 10623 Berlin, Germany. [2]Center for Theoretical Physics and School of Science, Hainan University, Haikou 570228, China. [3]Department of Physics and Research Center OPTIMAS, University Kaiserslautern-Landau, 67663 Kaiserslautern, Germany. [4]These authors contributed equally: Ling-Na Wu, Jens Nettersheim. ✉e-mail: eckardt@tu-berlin.de; widera@rptu.de

behaviour of driven-dissipative quantum systems in response to control parameters other than time (see, e.g., ref. 21).

In contrast to isolated quantum systems, open quantum systems[22] are characterized by the coupling to an environment, called a bath, with which they exchange both energy and information. Markovian baths rapidly dissipate information, so that the dynamics of the system can be described by an idealized time-local master equation $\dot{\rho}(t) = \mathcal{L}[\rho(t)]$, where the dynamics is generated by a time-independent Liouvillian superoperator $\mathcal{L}$ acting on the instantaneous density operator $\rho(t)$ describing the system's state at time $t$. If the coupling to the environment is weak compared to the level spacing in the system, $\rho(t)$ quickly becomes diagonal with respect to the energy eigenstates $|m\rangle$, $\rho(t) \simeq \sum_m p_m(t)|m\rangle\langle m|$. The probabilities $p_m(t)$ for being in state $|m\rangle$ then follow a Pauli rate equation $\dot{p}_m = \sum_{m' \neq m}[R_{mm'}p_{m'} - R_{m'm}p_m]$, with $R_{mm'}$ denoting the rate for a bath-induced transition from $|m'\rangle$ to $|m\rangle$[22].

## Results

In systems of ultracold atoms, dissipation can be engineered in various ways, including, for instance, the coupling of the atoms to a cavity[23], spontaneous emission of lattice photons[24,25], particle loss (e.g. via controlled ionization[26]), or the coupling to a background gas[27]. We realize such an open system by the spin degrees of freedom of individual ultracold Caesium atoms ($^{133}$Cs), which are immersed as impurities in a bath comprising ultracold Rubidium atoms ($^{87}$Rb) [see the sketch in Fig. 1a and Methods for details]. The hyperfine states of both species form stable quasi-spins with quantum numbers $F = 3$ ($F = 1$) for Cs (Rb). In the presence of a weak, constant external magnetic field $B$, the spins possess an equidistant ladder spectrum $E_{m_F} = m_F\Delta$, where $\Delta = g_F\mu_B B/\hbar$, with Landé factor $g_F$, reduced Planck constant $\hbar$ and Bohr magneton $\mu_B$. The corresponding energy eigenstates $|m_F\rangle$ are characterized by the magnetic quantum number $m_F = -F, -F+1, ..., F$. While the Cs $|m_F = 3\rangle$ state is the ground state of the isolated Cs atom, it is the highest excited Cs state of the open Cs-Rb system. Controlling the initial Cs-state population allows experimentally initializing the open-system dynamics with almost arbitrary excitation energy of the Cs spin. Elastic Rb-Cs collisions quickly thermalize the Cs atoms' center-of-mass motion, while inelastic spin-exchange (SE) processes give rise to bath-induced transitions, where the Cs spins are changed by single quanta of angular momentum, $m_F \rightarrow m'_F = m_F \pm 1$ with corresponding

rates $R_{\pm,m_F} \equiv R_{m_F \pm 1,m_F}$[28], see Fig. 1b and Methods ($m_F$ is used throughout for the Cs spins). The combination of a large atom-number imbalance, the ratio of elastic to inelastic collision rates, and a relatively large mean-free path realize an almost ideal Markov bath, yielding a collision probability of a Cs impurity with the same Rb atom of well below a percent (see Supplementary Information Note 1). We initially prepare the Cs impurity in an excited spin state defined by the probability distribution $p_{m_F}(0)$, and monitor the subsequent relaxation dynamics $p_{m_F}(t)$; see Fig. 1c for an example. As an important observable, we extract the evolution of the total entropy of our spin system [blue curve in Fig. 1c],

$$S(t) = -\sum_{m_F} p_{m_F}(t)\ln(p_{m_F}(t)). \tag{1}$$

In Fig. 2a and b, we show the measured evolution of $S$ for various different initial conditions (specified in the insets). This paper's blue and red background colors indicate unidirectional and bidirectional spin-exchange. For highly excited initial states, that means for the Rb-Cs compound states of large positive $m_F$ (for more details, see Methods), we find in both scenarios that the entropy evolution is highly non-monotonous. The entropy first increases to reach a peak value $S_{peak}$ at a time $t_{peak}$, before eventually relaxing to a steady-state. This is remarkable and rather different from the behavior found for initial states close to equilibrium, for which we observe that the entropy simply increases in time until it saturates at its steady-state value [see pink curve in Fig. 2b]. Even more remarkably, for various different initial conditions, i.e., different $m_F$-states and their combinations (see Supplementary Figs. 1 and 2 for a discussion of the role of the initial state and its energy), this peak value almost reaches the maximal possible entropy, $S_{max} = \ln M \approx 1.95$ with $M = 7$ being the number of spin states, indicated by the dashed line.

Experimentally, the total signal of $m_F$ populations as well as of the entropy value is the average over Cs-atom signals locally interacting with the inhomogeneous density distribution of the bath. Importantly, since the Cs impurities undergo approximately ten elastic collisions between two spin-exchange collisions and, moreover, the Cs mean-free path in the Rb cloud is of the order of the Rb cloud's extension (see Supplementary Note 1), each Cs atom samples the whole

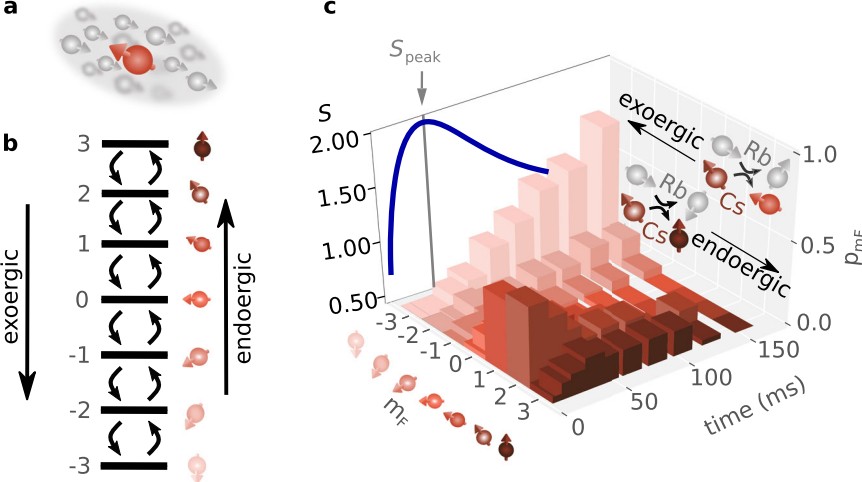

**Fig. 1 | Realizing an open spin system. a** Individual Cs atom (red) interacting with a bath of spin-polarized Rb atoms via inelastic spin-exchange (SE) collisions. **b** The Cs-Zeeman states experimentally realize an equidistant seven-level ($m_F = 3, 2, ..., -3$) spin system defining $m_F = -3$ as ground state for the Rb-Cs compound (for more details, see Methods). SE collisions with the Rb atoms give rise to dissipative spin dynamics, increasing (decreasing) internal energy and angular momentum for endoergic (exoergic) processes. The twelve SE rates between the Cs Zeeman states depend on the external magnetic field and the bath temperature. **c** Bath-driven and time-resolved quantum-spin evolution for individual Cs atoms initially prepared in a mixture of $|m_F = 1\rangle$ and $|m_F = 2\rangle$. Sketches in the back panel show the microscopic collision processes of exoergic and endoergic SE collisions. The lateral plane shows the resulting entropy evolution, featuring a maximum at $S_{peak} = 1.944 \approx \ln 7 = S_{max}$.

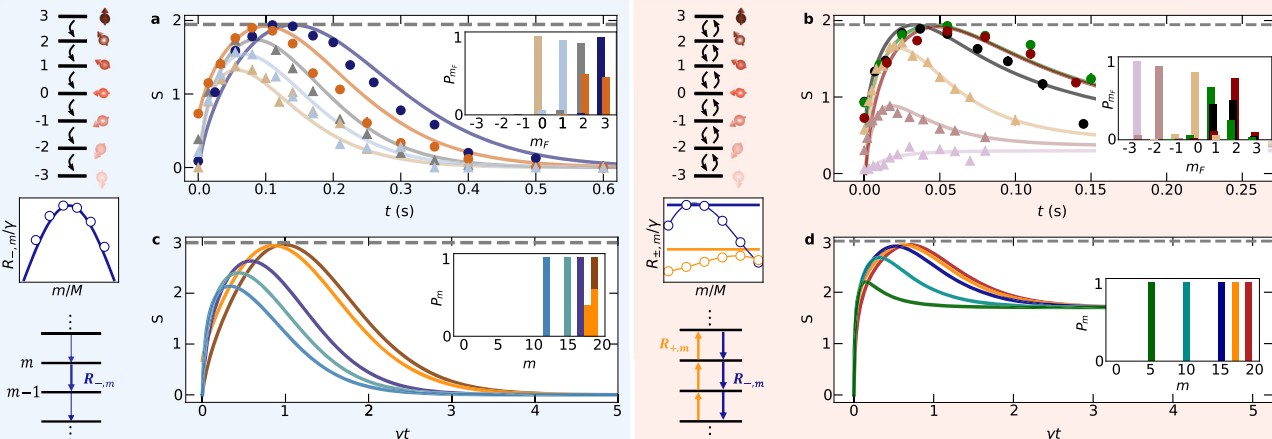

**Fig. 2 | Entropy evolution.** Blue (red) background indicates regimes where unidirectional spin transitions, applying a high magnetic field ($B = 460$ mG) and bidirectional spin transitions, utilizing a low magnetic field ($B = 25$ mG) are possible. While in the former case, endoergic transitions are suppressed completely[33], in the latter case, they are allowed and raise $m_F$ for Cs, but with reduced probability compared to exoergic processes, which lower $m_F$. **a, b** Experimentally measured entropy evolution starting from different initial states shown in the insets (colors match). Bullets (●) are used for trajectories with high peak entropy ($S_{peak} \geq 0.98 S_{max}$) and triangles (▲) otherwise. Error bars representing $1\sigma$ standard deviation statistical fluctuations are smaller than the symbol size. Solid lines are obtained from simulations, dashed lines indicate maximum possible entropy $S_{max}$. **c, d** Like **a, b** but for theoretical models with 20 states. Panels between the level schemes show normalized transfer rates for the experimental system (dots) and the theoretical models (lines). For the unidirectional model (in blue background), $R_{-,m} = \gamma M \sin(\pi(m+1)/M)$, where $\gamma$ is the coupling strength. For the bidirectional model (red background), thick horizontal lines correspond to the state-independent-rate model, with $R_{+,m} \equiv R_+ = \gamma M$ (orange) and $R_{-,m} \equiv R_- = \gamma M \exp(10/M)$ (blue).

inhomogeneous density during its relaxation. Therefore, the spin-population dynamics reflects a homogeneous broadening rather than local dynamics. Furthermore, since the number of $m_F$ states is bound for the Cs impurity, an average can only lower the total entropy signal, so that the measured values close to the maximum value are a lower bound.

## Discussion

The approach of $S_{max}$ implies that the system transiently approaches the maximally mixed state $\rho_{max} = M^{-1} \sum_{m_F} |m_F\rangle\langle m_F|$, corresponding to a completely delocalized spin distribution $p_{m_F} = 1/M$. In the limit of large $M$, such behavior implies a divergence of both $S$ and the "length" $\xi$ that characterizes the number of states $|m_F\rangle$ covered by the probability distribution $p_{m_F}$. The latter can, e.g., be defined as the participation ratio, $\xi \equiv (\sum_{m_F} p_{m_F}^2)^{-1}$.

This, in turn, directly corresponds to the divergence of a relevant length scale $\xi$ that is found when a system approaches a critical point like at a continuous phase transition. Here, however, the continuous control parameter is the time $t$ and its critical value is $t_{peak}$. In this sense, a transient approach of the maximally mixed state $\rho_{max}$ resembles the behavior of a phase transition in time in the limit of large $M$.

To answer the question whether the observed dynamics is indeed a finite-size precursor of a phase transition, we define two model systems of variable system size $M$ [see Fig. 2 middle and lower side panels] and numerically perform a finite-size scaling analysis to extract the behavior for $M \to \infty$. Both models consist of $M$ states labeled by $m = 0$, $1,...,M-1$, which form an equidistant energy spectrum $\varepsilon_m = m\Delta$. A unidirectional model generalizes the high-magnetic-field regime to larger $M$. Here, only transitions $|m\rangle \to |m' = m-1\rangle$ occur, corresponding to a zero-temperature bath. The rates $R_{-,m}$ possess a parabolic dependence on $m$ mimicking the experimental rates. Such a rate inhomogeneity is required for reaching high peak entropies, since for unidirectional transport a right-moving probability distribution can only become broader if the velocity at right end is larger than at its left end. In a bidirectional model, corresponding to the case of low magnetic fields, it is sufficient to assume state independent rates. Figure 2c, d depict the entropy evolution for both models with $M = 20$

for various initial conditions. Again $S_{peak} \approx S_{max}$ is found for highly excited initial states.

To compare data for different initial conditions, we introduce the scaled control parameter

$$\beta_{eff} \equiv \frac{dS/dt}{dE/dt} = \frac{dS}{dE}, \tag{2}$$

with mean energy $E$, having the dimension of an inverse temperature. It is monotonically related to the time $t$ (see Supplementary Fig. 3 for more details) and becomes zero at $t = t_{peak}$, while it takes negative (positive) values for $t < t_{peak}$ ($t > t_{peak}$). Despite superficially resembling an effective inverse temperature, we would like to stress that the use of this parameter does not imply that the system assumes a Gibbs-like state with effective time-dependent inverse temperature $\beta_{eff}$ during its transient evolution. In Fig. 3a, c, the measured entropy is plotted as a function of $\beta_{eff}$. For those initial conditions giving rise to close-to-maximum peak entropies, marked by bullets, the data collapse in the vicinity of $\beta_{eff} = 0$. Such behavior is equally visible when plotting the scaled localization length $\xi/M$ [Fig. 3b, d]. The insets show results for initial conditions corresponding to less excited states, for which the data does not collapse.

Assuming a continuous phase transition in the thermodynamic limit, $M \to \infty$, the localization length in this limit, $\xi_\infty$, is expected to diverge like $\xi_\infty \propto \beta_{eff}^{-\nu}$ at the transition point $\beta_{eff} = 0$, where $\nu$ is a critical exponent. As a consequence, the behavior of a large finite system should asymptotically depend on the ratio of $\xi_\infty$ and the system size $M$ only, or, equivalently on $(\xi_\infty/M)^{-1/\nu} = \beta_{eff} M^{1/\nu}$ [1]. In particular, the length $\xi$ for a system of finite large size $M$ is expected to behave as $\xi/M = g(\beta_{eff} M^{1/\nu})$ close to the transition point $\beta_{eff} = 0$, with some scaling function $g$. In Fig. 4 we show $\xi/M$ for different system sizes $M$ as a function of both the dimensionless control parameter $\Delta\beta_{eff}$ as well as the scaled control parameter $\Delta\beta_{eff} M$, corresponding to a critical exponent of $\nu = 1$. Remarkably, in the latter case, we find an almost perfect collapse of the data, suggesting universal scaling as it is found at a continuous phase transition. We note that the peak of $\xi/M$ does not fully reach 1. However, since the maximum of $\xi/M$ remains constant with increasing system size, $\xi$ diverges at $t_{peak}$ in the thermodynamic

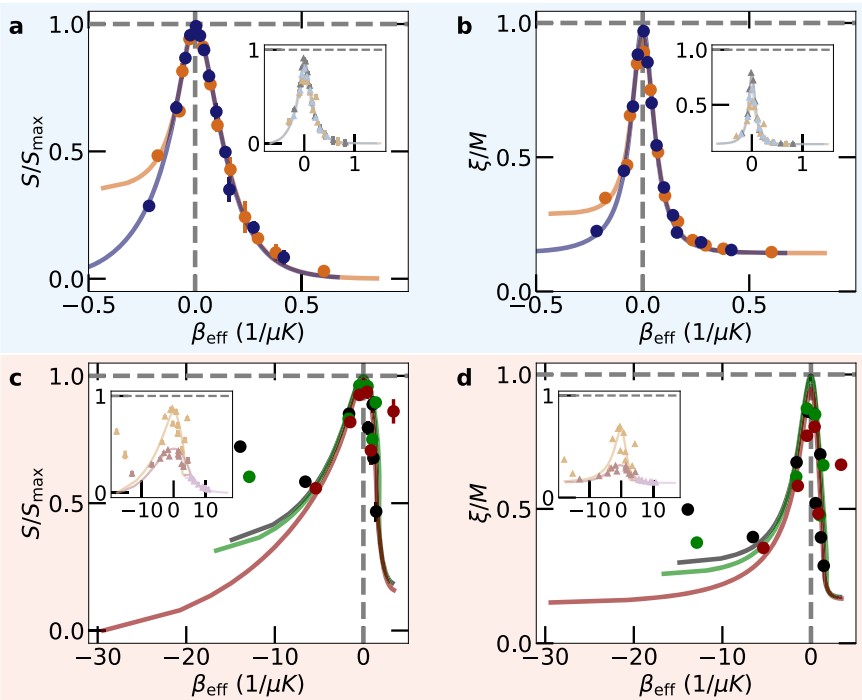

**Fig. 3 | Control parameter.** Experimentally measured (symbols) and simulated (lines) entropy (**a**, **c**) and localization length (**b**, **d**) plotted as a function of $\beta_{eff}$ [data, colors, and symbols like in Fig. 2a, b]. The main (inset) panel shows the results for the initial conditions far from (close to) equilibrium. Error bars represent statistical fluctuations of $1\sigma$ standard deviation. The error bars for $\xi/M$ are too small to be seen.

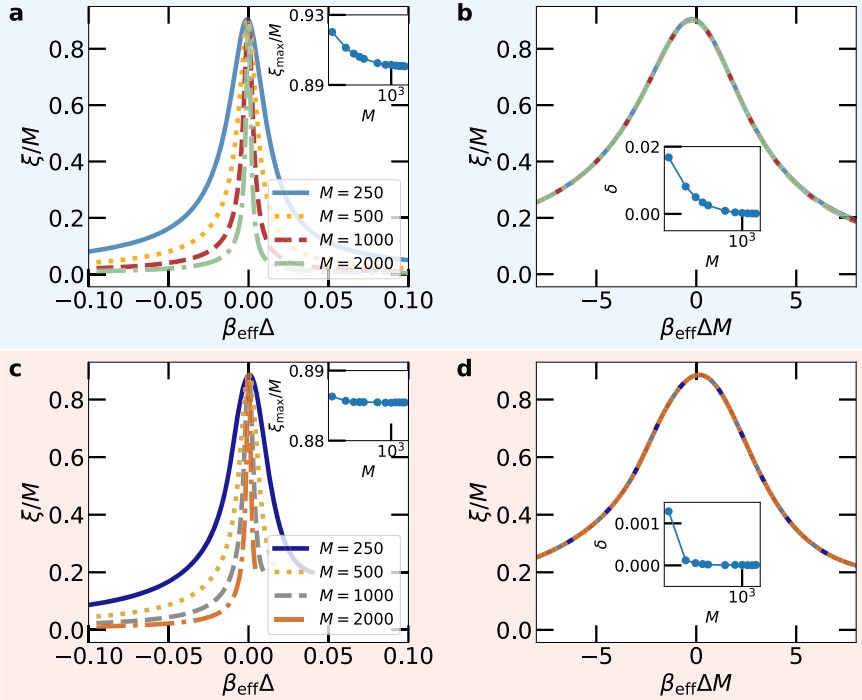

**Fig. 4 | Finite-size scaling.** Scaled Localization length $\xi/M$ for different system sizes $M$ for the unidirectional (**a**, **b**) and the bidirectional (**c**, **d**) model as a function of $\beta_{eff}\Delta$ (**a**, **c**) and the scaled parameter $\beta_{eff}\Delta M$ corresponding to $\nu = 1$ (**b**, **d**). The insets show the maximal $\xi$ as a function of the system size and the mean distance $\delta$ of the data for system size $M$ with that for system size $M = 2000$.

limit, similar to the critical behavior found at a continuous phase transition.

The universal behaviour of the dynamics observed in the large-system limit can be understood better by mapping the thermodynamic limit to a continuum limit: For a hypothetical system of fixed size $L$, the variable $x = Lm/M$ becomes continuous for $M \to \infty$ and the rate equation for the probability distribution $p_m$ approaches a differential equation for the probability density $\rho(x) = (M/L)p_{xM/L}$. Close to the transition, the observed exponent $\nu = 1$ can then be explained by starting from a maximally delocalized distribution, $\rho(x) = 1/L$, and

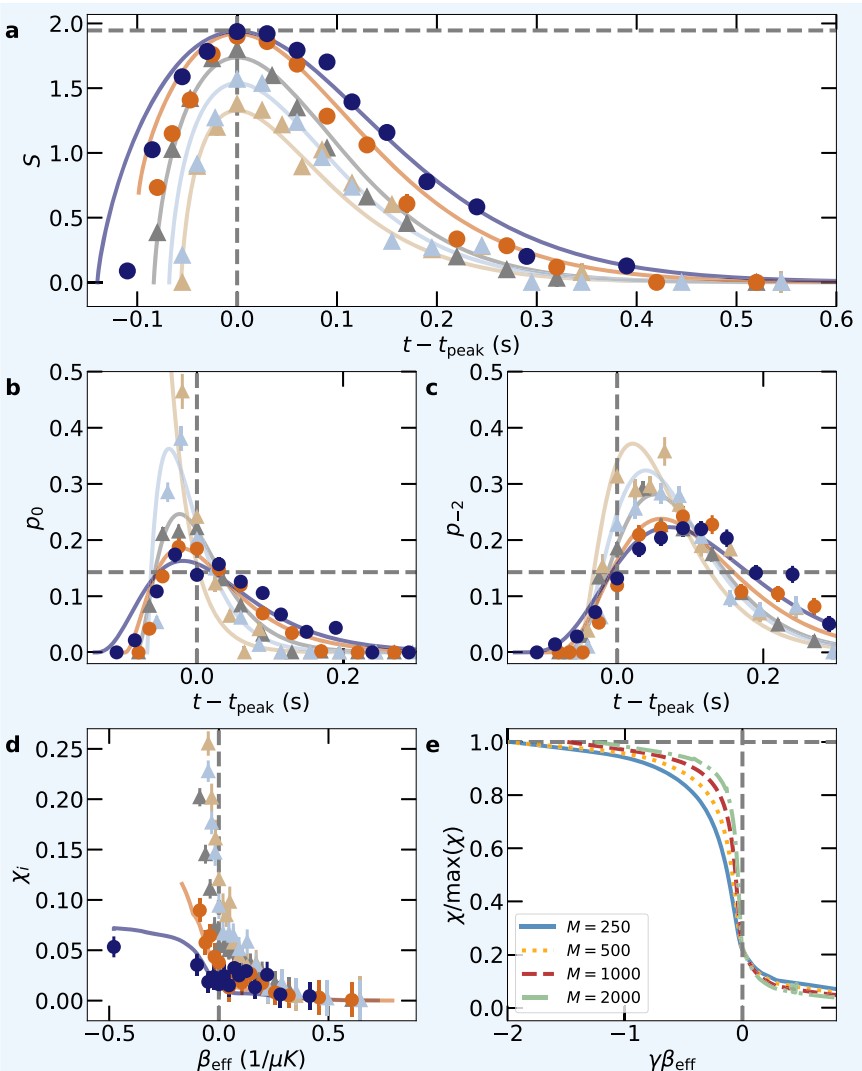

**Fig. 5 | Prethermal memory loss. a** Experimentally measured (symbols) and simulated (lines) entropy for different initial conditions (color coding as in Fig. 2a) and **b**, **c** the corresponding population $p_{m_F}$ for the two spin states (**b**) $|m_F = 0\rangle$ and (**c**) $|m_F = -2\rangle$ as a function of shifted time (by the peak entropy time $t_{peak}$) for the unidirectional model. The dependency of $\chi$ (see text for the definition) on the control parameter $\beta_{eff}$ is shown in **d** for the experiments and in **e** for the theoretical model. Horizontal grey dashed lines mark the maximal entropy $S_{max}$ in **a**, the population 1/7 in **b**, **c** that corresponds to $S_{max}$, and 1 in **e**. Vertical dashed lines mark $t = t_{peak}$ and $\beta_{eff} = 0$, respectively. Error bars represent statistical fluctuations of $1\sigma$ standard deviation.

studying the evolution for small positive and negative times perturbatively. More details can be found in Supplementary Note 3 and Supplementary Figs. 7 and 8), where we also present further analyses, showing universal scaling of the specific-heat-like quantity $C = dE/d(\beta_{eff}^{-1})$, and discussing the phase transition with time $t$ (rather than $\beta_{eff}$) playing the role of the control parameter (Supplementary Fig. 4).

Since the maximum entropy $S_{max}$ corresponds to a unique state, the maximally mixed state $\rho_{max}$, after approaching $S \approx S_{max}$, the dynamics is expected to become (approximately) independent of the details of the initial state. This happens long before the spin system has thermalized[29]. Such a prethermal memory loss is observed in the experiment for both magnetic field regimes. In Fig. 5a–c, we show the entropy evolution and the population dynamics of two spin states versus the shifted time $t-t_{peak}$ in the regime of unidirectional rates (corresponding plots for the bidirectional regime are presented in Supplementary Fig. 5). We can see that the data with a high peak entropy ($S_{peak} \geq 0.98 S_{max}$, indicated by bullets) show similar behavior for both the entropy evolution and spin dynamics after the system reaches peak entropy ($\tau \equiv t - t_{peak} > 0$).

To quantify this observation, we introduce the distance

$$\chi_{ij}(\tau) = \frac{1}{M}\sum_{m=1}^{M}\left|p_m^{(i)}(t_{peak}^{(i)} + \tau) - p_m^{(j)}(t_{peak}^{(j)} + \tau)\right| \quad (3)$$

between two trajectories with different initial conditions, $p_{m_F}^{(i)}(0)$ and $p_{m_F}^{(j)}(0)$, and peak times $t_{peak}^{(i)}$ and $t_{peak}^{(j)}$. For the experimental data we compare each trajectory $p_{m_F}^{(i)}$ to the optimal trajectory $p_{m_F}^{max}(t)$ defined by $S_{peak} = S_{max}$. In Fig. 5d, we plot the corresponding distance $\chi_i \equiv \chi_{i\,max}$ versus $\beta_{eff}$. For those trajectories featuring large peak entropies, $\chi_i$ becomes small at the transition $\beta_{eff} = 0$. In comparison, for trajectories with $S_{peak} < 0.98 S_{max}$ (indicated by triangles) $\chi_i$ remains large after the transition.

Prethermal memory loss is also found in the theoretical models (see Supplementary Fig. 6). Here we have easy access to many initial conditions and, therefore, we can compute the mean distance $\chi \equiv \text{mean}_{ij \in \mathcal{U}}(\chi_{ij})$ of those trajectories whose peak entropies $S_{peak}$ are close to the maximal entropy, i.e., for which $S_{peak}/S_{max} > 1 - \delta S$ with

threshold $\delta S \ll 1$. Fig. 5e plots normalized $\chi$ versus $\beta_{\text{eff}}$ for different system sizes with $\delta S = 0.2$. One can see that for increasing $M$ a sharp transition forms at $\beta_{\text{eff}} = 0$.

In summary, we have investigated the far-from-equilibrium relaxation dynamics of an open quantum system given by a large spin coupled to a bath. We find that for highly excited initial states, the system transiently approaches the maximally mixed state $\rho_{\max}$, as signaled by a peak in the entropy evolution approximately reaching the maximally possible value $S_{\max}$. We show that, when reaching the entropy peak, the dynamics shows distinct features that signal critical scaling with respect to time: (i) In the limit of large system sizes, the localization length $\xi$ characterizing the spin state, diverges at the transition. (ii) A finite-size scaling analysis reveals a power-law scaling $\xi \sim \beta_{\text{eff}}^{-\nu}$ near the transition, with respect to the scaled control parameter $\beta_{\text{eff}}$, which is monotonically related to time and allows to compare data for different initial states by locating the transition to $\beta_{\text{eff}} = 0$. (iii) The extracted critical exponent takes the same value $\nu = 1$ for all model parameters considered, suggesting universal scaling behavior independent of the microscopic details of the system. Thus, we conclude that critical behavior with respect to time can not only occur in the evolution of isolated systems described by pure states but also during the dynamics of an open system induced by dissipation. It will be interesting to further investigate the nature of such dynamical critical scaling in open quantum systems, including its non-equilibrium universality classes (to the exploration of which our results provide a first step and a new approach). Another subject for future theoretical and experimental exploration is the collective behaviour of many atoms in contact with the bath as it results both from quantum statistics as well as from potential interactions. Also, the regime of stronger system-bath coupling, where non-markovian effects are expected, offers an intriguing perspective.

## Methods
### Initial state preparation
Experimentally, the Rb bath is prepared by laser-cooling in a magneto-optical trap (MOT) and subsequent cooling by evaporation while the sample is trapped in a crossed dipole trap at a wavelength of $\lambda = 1064$ nm. The bath's internal state is prepared via an optical pumping in $|F_{\text{Rb}} = 1, m_{F,\text{Rb}} = 1\rangle$ and then transferred via the radio-frequency transition $|F_{\text{Rb}} = 1, m_{F,\text{Rb}} = 1\rangle \rightarrow |F_{\text{Rb}} = 1, m_{F,\text{Rb}} = 0\rangle$ to a magnetic-field insensitive state. This allows us to accumulate Cs atoms from the atomic background vapor by laser cooling in a MOT only approximately $200\mu m$ apart from the Rb sample. Subsequently, a crossed dipole trap with a wavelength of $\lambda$ loads the atoms from the MOT. Degenerate Raman sideband cooling[30] reduces the Cs temperature further while at the same time populating the bare atoms' absolute ground state. Microwave-driven Landau-Zener transitions near-resonant to the $|F_{\text{Cs}} = 3\rangle \rightarrow |F_{\text{Cs}} = 4\rangle$ hyperfine transition ($h \times 9.1$GHz) prepare the Cs atoms in the desired initial state.

The interaction between Cs and Rb is initialized by transporting the Cs atoms into the bath via a species-selective optical lattice[31]. The interaction stops after applying a resonant laser pulse that pushes the Rb atoms out of the trap. Eventually, state-selective fluorescence imaging[32] yields the internal state and position of the Cs atoms.

### Experimental parameters
The bath temperature $T$ and density $n$ for each measurement are inferred from time-of-flight measurements of the Rb cloud on the one hand; and from comparing the seven measured $m_F$-state trajectories with hundreds of simulated state trajectories on the other hand. Each simulation contains slightly different bath parameters. The bath parameters yielding the smallest least-squares ($\chi^2$) error for all trajectories and the independent time-of-flight measurement was used for the respective measurement data set. The individual parameters of

## Table 1 | Experimental parameters (magnetic field $B$, temperature $T$, atom density $n$) and ratio of mean rates $R_{+,m_F}/R_{-,m_F}$.

| Parameter | Unidirectional | Bidirectional |
|---|---|---|
| $B$ [mG] | $460 \pm 2$ | $25 \pm 2$ |
| $T$ [nK] | $920 \pm 24$ | $492 \pm 31$ |
| $n$ [$10^{13}$cm$^{-3}$] | $0.46 \pm 0.02$ | $0.51 \pm 0.09$ |
| $R_{+,m_F}/R_{-,m_F}$ | $\approx 10^{-5}$ | $0.21$ |

each measurement and the corresponding initial population are listed in the Supplementary Tables 1 and 2. For simplicity, Table 1 shows the mean temperature and mean density of all best-fitting parameters for the unidirectional, respectively, bidirectional experimental system. Moreover, the magnetic field is calibrated via microwave spectroscopy on the $|F_{\text{Rb}} = 1, m_{F,\text{Rb}} = 0\rangle \rightarrow |F_{\text{Rb}} = 2, m_{F,\text{Rb}} = 1\rangle$ transition of the Rb bath.

### Inter-species spin-exchange processes
The Zeeman energy for a bare Cs atom reaches its minimum for $|m_F = 3\rangle$, defining the single-atom ground state. However, the situation reverses when the Cs atom is immersed in a bath of Rb atoms in the $|m_{F,\text{Rb}} = 0\rangle$ state. For this Rb-Cs combination, spin-exchange collisions can exchange one quantum of angular momentum between one atom of the bath and the Cs atom while the total angular momentum is preserved. At the same time, Zeeman energy is exchanged. Due to different atomic Landé factors, the Zeeman splitting of Rb is twice the splitting of Cs. Therefore, the spin- and energy exchange direction is essential and corresponds to two complementary processes in the bath. The process $|m_F^{\text{Cs}}, m_F^{\text{Rb}}\rangle \rightarrow |m_F^{\text{Cs}} - 1, m_F^{\text{Rb}} + 1\rangle$ is exoergic, and the energy amount corresponding to one Cs atom's Zeeman energy $\hbar\Delta = \mu_B g_F^{\text{Cs}} B$ is released as kinetic energy and dissipated by subsequent elastic collisions in the bath. The complementary process $|m_F^{\text{Cs}}, m_F^{\text{Rb}}\rangle \rightarrow |m_F^{\text{Cs}} + 1, m_F^{\text{Rb}} - 1\rangle$ is endoergic, and the kinetic collisional energy of the Cs atom and bath atom must provide the energy amount $\hbar\Delta$ for this collision to occur. The collisional energy is Maxwell-Boltzmann distributed. For the ultracold temperatures of the bath, the rates for exothermal and endothermal SE collisions, $R_-$ and $R_+$, respectively, have markedly different rates with $R_- > R_+$. As a consequence, the definitions of ground and highest excited states invert, and the former bare-atom ground (highest-excited) state, i.e., $|m_F = +3\rangle$ ($|m_F = -3\rangle$), defines the impurity's highest excited (ground) state.

### Spin evolution calculation
The evolution of the probability in eigenstate $|m\rangle$, $p_m$, is described by the rate equation

$$\dot{p}_m = R_{+,m-1} p_{m-1} + R_{-,m+1} p_{m+1} - (R_{-,m} + R_{+,m}) p_m. \qquad (4)$$

where $R_{\pm,m} \equiv R_{m\pm 1,m}$ denotes the transfer rate from eigenstate $|m\rangle$ to eigenstate $|m \pm 1\rangle$. For the unidirectional model discussed in the main text, $R_{+,m} = 0$. For the bidirectional model with state-independent rates, $R_{\pm,m} \equiv R_{\pm}$.

For simulating the experimental spin dynamic, the rates are given by $R_i = \langle n\rangle \sigma_i(B,T)\bar{v}$, with $i = m_F \pm 1, m_F$, mean relative velocity of the colliding atoms $\bar{v}$, Cs-Rb density overlap $\langle n\rangle$ and state-dependent scattering crossing section $\sigma_i$. The ratio of the mean rates $R_{+,m_F}/R_{-,m_F}$ in Table 1 shows an experimentally accurately blocking of the endothermal rates $R_{+,m_F}$ by choice of a large magnetic field.

## Data availability
All data supporting the finding of this paper are available in a Zenodo repository: 10.5281/zenodo.10526596.

## Code availability

The codes that support the findings of this paper are available from the corresponding author A.E. upon request.

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

## Acknowledgements

We thank Markus Heyl, Eric Lutz, and Li You, for their useful comments on the manuscript. This work was supported by the Deutsche Forschungsgemeinschaft (DFG, German Research Foundation) via the Collaborative Research Centers SFB/TR185 (Project No. 277625399) and SFB 910 (Project No. 163436311). S.B. acknowledges funding from Studienstiftung des deutschen Volkes.

## Author contributions

A.W. and A.E. conceived the project. J.N., J.F., S.B., S.H., and D.A. run the experimental apparatus and took the data. J.N., L.W., and J.F. analyzed the data. L.W. and A.S. developed the theoretical models. All authors contributed to the interpretation of the data and writing of the manuscript.

## Funding

## Competing interests

The authors declare no competing interests.
