## [Peer Review File · Nature Communications]

Indication of critical scaling in time during the relaxation of an open quantum systemEditorial Note: This manuscript has been previously reviewed at another journal that is not operating a transparent peer review scheme. This document only contains reviewer comments and rebuttal letters for versions considered at *Nature Communications*.

REVIEWERS' COMMENTS

Reviewer #1 (Remarks to the Author):

In the reply letter, the author claims two non-trivial aspects of this work. First, during relaxation dynamics of the Caesium atoms, the entropy $S(t)$ of the atomic spin state follows thermal description with effective temperature. Second, during the relaxation dynamics, the entropy reaches a maximum with scaling behaviour. However, if the Caesium atoms can interact with its surrounding Rubidium bath, it is not super surprising that the spin dynamics follow the thermal distribution. One can also expect that it requires high energy initial state to reach maximal entropy as shown in the Fig. 2 and Fig. S5. Moreover, the scaling argument is mainly supported by numerical studies with a high spin number, which is the classical limit.

In some parts, I do agree that the relaxation dynamics of the atomic spin state of a single atom itself could be interesting because this is the first demonstration of the single-atom thermalization process. So, this work may be worthy of publication in *Nature Communications*. However, in my opinion, there are no special reasons for interpreting the results as a phase transition in an open quantum system. What are the new perspectives that can be seen in this view point?

Before the recommendation of the publication, the author should revise the manuscript so as to avoid placing too much focus on phase transition. The abstract and introduction are still pointing out that the experiment shows a phase transition in an open system. Not surprisingly, one can also see the claim that the result indicates the phase transition in the main text. E.g., page 3: "... indicates a phase transition in time in the limit of large M ". page 4, "... These observations suggest the interpretation as a phase transition with respect to time."

Response to Referees Letter

REVIEWERS' COMMENTS

Reviewer #1 (Remarks to the Author):

In the reply letter, the author claims two non-trivial aspects of this work. First, during relaxation dynamics of the Caesium atoms, the entropy $S(t)$ of the atomic spin state follows thermal description with effective temperature. Second, during the relaxation dynamics, the entropy reaches a maximum with scaling behaviour. However, if the Caesium atoms can interact with its surrounding Rubidium bath, it is not super surprising that the spin dynamics follow the thermal distribution. One can also expect that it requires high energy initial state to reach maximal entropy as shown in the Fig. 2 and Fig. S5. Moreover, the scaling argument is mainly supported by numerical studies with a high spin number, which is the classical limit.

In some parts, I do agree that the relaxation dynamics of the atomic spin state of a single atom itself could be interesting because this is the first demonstration of the single-atom thermalization process. So, this work may be worthy of publication in Nature Communications. However, in my opinion, there are no special reasons for interpreting the results as a phase transition in an open quantum system. What are the new perspectives that can be seen in this view point?

Before the recommendation of the publication, the author should revise the manuscript so as to avoid placing too much focus on phase transition. The abstract and introduction are still pointing out that the experiment shows a phase transition in an open system. Not surprisingly, one can also see the claim that the result indicates the phase transition in the main text. E.g., page 3: "... indicates a phase transition in time in the limit of large M ". page 4, "... These observations suggest the interpretation as a phase transition with respect to time."

Author response

We thank the reviewer for their work and comments on our manuscript.

The reviewer writes that the observation of a temperature-like distribution is not too surprising. In contrast, we emphasize that this is not expected at all because, first, the system does not interact and

therefore no eigenstate thermalization can take place; and second, the thermal bath has a temperature that is always different from the effective spin temperature, except for asymptotically long times.

Although the critical exponents are determined numerically, the numerical model is firmly anchored by the quantitative agreement with the experimental values. The qualitative properties are the same, only for quantitative determinations in the thermodynamic limit the numerical model was used.

As for the requested modifications, we have been careful not to claim the observation of a phase transition and we have modified the manuscript accordingly. However, we believe that the observation of critical scaling in time and universal exponents is closely related to the phenomenon of phase transitions, and we think that the discussion of critical scaling in a physical system requires the introduction of the connection to phase transitions.